# The Opioid Receptor Influences Circadian Rhythms in Human Keratinocytes through the β-Arrestin Pathway

**DOI:** 10.3390/cells13030232

**Published:** 2024-01-25

**Authors:** Paul Bigliardi, Seetanshu Junnarkar, Chinmay Markale, Sydney Lo, Elena Bigliardi, Alex Kalyuzhny, Sheena Ong, Ray Dunn, Walter Wahli, Mei Bigliardi-Qi

**Affiliations:** 1Department of Dermatology, University of Minnesota, Minneapolis, MN 55455, USAcmarkale@umn.edu (C.M.);; 2Stem Cell Institue, McGuire Translational Research Facility, University of Minnesota, Minneapolis, MN 55455, USA; 3Agency for Science, Technology and Research, Singapore 138632, Singapore; seetansh001@e.ntu.edu.sg (S.J.); sheenaonglm@gmail.com (S.O.); ray.dunn@ntu.edu.sg (R.D.); 4Department of Neuroscience, Medical School, University of Minnesota, Minneapolis, MN 55455, USA; kalyu001@umn.edu; 5Lee Kong Chian School of Medicine, Nanyang Technological University, Singapore 639798, Singapore; walter.wahli@ntu.edu.sg; 6Unité Mixte de Recherche (UMR) 1331, Institut National de la Recherche Agronomique (INRA), ToxAlim, 31000 Toulouse, France; 7Center for Integrative Genomics, University of Lausanne, 1015 Lausanne, Switzerland

**Keywords:** opioid receptor, circadian rhythm, keratinocyte, beta-arrestin

## Abstract

The recent emphasis on circadian rhythmicity in critical skin cell functions related to homeostasis, regeneration and aging has shed light on the importance of the *PER2* circadian clock gene as a vital antitumor gene. Furthermore, delta-opioid receptors (DOPrs) have been identified as playing a crucial role in skin differentiation, proliferation and migration, which are not only essential for wound healing but also contribute to cancer development. In this study, we propose a significant association between cutaneous opioid receptor (OPr) activity and circadian rhythmicity. To investigate this link, we conducted a 48 h circadian rhythm experiment, during which RNA samples were collected every 5 h. We discovered that the activation of DOPr by its endogenous agonist Met-Enkephalin in N/TERT-1 keratinocytes, synchronized by dexamethasone, resulted in a statistically significant 5.6 h delay in the expression of the core clock gene *PER2*. Confocal microscopy further confirmed the simultaneous nuclear localization of the DOPr-β-arrestin-1 complex. Additionally, DOPr activation not only enhanced but also induced a phase shift in the rhythmic binding of β-arrestin-1 to the *PER2* promoter. Furthermore, we observed that β-arrestin-1 regulates the transcription of its target genes, including *PER2*, by facilitating histone-4 acetylation. Through the ChIP assay, we determined that Met-Enkephalin enhances β-arrestin-1 binding to acetylated H4 in the *PER2* promoter. In summary, our findings suggest that DOPr activation leads to a phase shift in *PER2* expression via β-arrestin-1-facilitated chromatin remodeling. Consequently, these results indicate that DOPr, much like its role in wound healing, may also play a part in cancer development by influencing *PER2*.

## 1. Introduction

Skin is exposed to various environmental and stress factors depending on the time of day. Studies attest to the existence of circadian rhythmicity in clock gene expression in keratinocytes, fibroblasts and melanocytes [1]. Gene expression profiling of mouse skin identified approximately 1400 genes with circadian expression patterns, implying that the circadian clock may influence several facets of skin physiology. An autoregulatory gene expression feedback loop forms the core of the circadian clock. The core loop’s positive limb consists of the Brain and muscle ARNT-like protein 1 (BMAL1) and Circadian locomotor output cycles kaput (CLOCK) proteins, while the negative limb comprises the Period (PER) and Cryptochrome (CRY) proteins [1].

Circadian rhythmicity is involved in various cutaneous cellular and physiological functions such as pain intensity perception, tissue homeostasis, regeneration, DNA repair mechanisms and aging [1,2,3,4]. Similar functions are mediated by the opioid receptor (OPr) system in skin, particularly in response to external and internal stressors [5]. Activation of peripheral OPrs via endogenous opioid peptides, including enkephalins, endorphins, dynorphins and endomorphins, can regulate various keratinocyte activities [6]. Delta opioid receptors (DOPrs) are known to play an important role in skin differentiation. Deletion of these receptors significantly delays wound healing [7]. 

It was previously reported that internalization of the DOPr occurs via the beta arrestin (βarr) pathway and that βarr1 becomes localized to the nucleus [8]. βarr are scaffolding proteins that bind to a G-protein coupled receptor, thus interrupting interactions between the receptor and its ligand [9]. βarr1 has been implicated as a molecular marker of peripheral circadian rhythm in human salivary glands; however, its interactions with cellular circadian components have not yet been elucidated [10]. 

Methionine-enkephalin (Met-ENK), primarily a Delta-OPr agonist, is processed from a precursor protein encoded by both the proenkephalin (PENK) and proopiomelanocortin (POMC) genes, which are expressed in human skin [11]. In the hamster central nervous system, OPr activity modulates circadian rhythms, wherein activation by a delta opioid agonist induces phase shifts in locomotor activity [12]. The inability of antagonists to modulate light-induced phase advances suggests that enkephalins are likely used by the circadian system to modulate responses only under certain conditions or time of day [13]. We sought to investigate whether a similar link could be established in skin cells. Specifically, we tested the effect of Met-ENK-induced OPr activation on the rhythmicity of clock gene expression in keratinocytes. 

Our focus is on the expression of the transcriptional activator BMAL1 and PER2, believed to mainly control phase advance and delay [14]. *PER2* is considered a main oscillatory gene and central pacemaker in mammals [15,16]. Induction of *PER2* mRNA has been observed to contribute to phase delay in mice [17], and Casein Kinase 1delta inhibition, a pharmacologic therapy used to adjust the circadian clock, has been shown to be dependent on endogenous *PER2* levels [18]. In recent years, studies have shown that the aberrant expression of *PER2* is responsible for not only circadian rhythm alterations but also the occurrence and development of cancers [19]. The loss of *PER2* expression is also associated with the development of oral and head and neck squamous cell carcinomas [20,21].

## 2. Materials and Methods

### 2.1. Cell Culture

Human skin keratinocyte cell line N/TERT-1 was obtained from and cultured as described by the Rheinwald Laboratory [22,23].

### 2.2. DNA Construct

The plasmid used for N/TERT-1 experiments was a lentiviral expression clone with pEZ-Lv122 vector backbone including the DOPr-GFP open reading frame (NCBI entry U10504) purchased from GeneCopoeia (no. EX-A1155-Lv122; Rockville, MD, USA). pReciever-Lv127 vector backbone including the β-arrestin 1—CFP open reading frame (NCBI Accession number NM_004041) was purchased from GeneCopoeia (no. EX-V0563-LV127; Rockville, MD, USA). SMARTvector lentiviral Human ARRB1 catalogue number V3SH11240-225279371 with an hCMV promoter was purchased from Dharmacon (Lafayette, CO, USA). The following oligonucleotides for the DOPr knockdown were used forward: 

5′-CGCGTCCCCTGCTCTCCATCGACTACTATTCAAGAGATAGTAGTCGATGGAGA GCATTTTTGGAA-3′ and reverse: 

5′-TTCCAAAAATGCTCTCCATCGACTACTATCTCTTGAATAGTAGTCGATGGAGAGCAGGGGA-3′. These oligonucleotide sequences were phosphorylated and inserted into dephosphorylated pLVTHM plasmids using a T4 ligase.

### 2.3. Treatments

Keratinocytes were seeded in 6-well plates or 10 cm dishes and used at 70% confluence for all experiments. On the day of the experiment, cells were treated with 1 μM dexamethasone (D4902; Sigma-Aldrich, Darmstadt, Germany) for one hour. Thereafter, the media were replaced with K-SFM (BPE- and EGF-free (for the control group)) or were treated with 100 nM Met-enkephalin (for the Met-ENK group) in keratinocyte serum-free medium (K-SFM) (bovine-pituitary-extract-free (BPE-free) and epidermal-growth-factor-free (EGF-free)) for 5 min. For examination of βarr, additional cell populations were treated with 10 μM Naltrindole (NTI) (for the NTI group) or 100 nM Met-ENK + 10 μM NTI (for the Met-ENK + NTI group). The knockdown DOPr (KD DOPr) keratinocytes (produced using a lentiviral construct described in the previous section) were similarly treated with 100 nM Met-ENK. All cells were kept under constant dark conditions in the incubator throughout the course of this experiment. Samples were collected every 5 h for the 45 h following synchronization with dexamethasone with no media changes. It was during this that we ran the following parallel experiments. 

### 2.4. RNA Isolation and Reverse-Transcription Polymerase Chain Reaction (RT-PCR)

Total RNA was extracted using RNeasy Kit (Qiagen, Hilden, Germany) according to the manufacturer’s protocol. Quality and concentration of the RNA in the samples were verified using a NanoDrop spectrophotometer. Total RNA (1 μg) was reverse-transcribed into cDNA in a 20 μL reaction using PrimeScript RT Reagent Kit (Takara, Japan). The resulting cDNA was diluted to obtain a concentration of 20 ng/μL cDNA.

### 2.5. Quantitative Real-Time Polymerase Chain Reaction (qPCR)

For the quantification of gene expression, 100 ng of cDNA template per reaction was amplified using SYBR green master mix and specific Quantitect primers from QIAGEN for *BMAL1, PER2, DEC2, DBP* and *TEF*. To test for the presence of DOPr expression, the following primers were used: forward: ACGTGCTTGTCATGTTCGGCATCGT, reverse: ATGGTGAGCGTGAAGATGCTGGTGA. Gene expression was normalized based on the values of the expression of RPL13A used as reference (forward primer 5′-CTC AAG GTC GTG CGT CTG AA-3′ and reverse primer 5′-TGG CTG TCA CTG CCT GGT ACT-3′). Quantification was performed using the comparative 2^−ΔΔCt^ method. For chromatin immunoprecipitation (ChIP) experiments discussed below, which were designed to test for the binding of βarr1 and DOPr to acetylated Histone 4 and CREB in the *PER2* promoter, the following target sequence was identified with the help of the UCSC genome browser. This sequence aligns with the promoter region of the human *PER2* gene.

5′-AGCCCCGGGAGGCCTGCATGCTGTTCACACACTCAGTCAGGTGGCCCCTCCTCCTGTTCCTCTGACATTGACACCTCGACACACTCCCCGCCCCCTCTCCCTAACACATACACACACAAATGCCAAGCAAACCCAGGCCCGGCTGCTGCGCCCTGCACACACCCAAAGGCTCTTTTGTTTCTTCCCTCCCATTGACGTCAATGGGGAGCTCCATTGTTCTGGAAACAAGAGTAAACAGACAGCTCATCCACACCTTACCGAGATTCTTCTTCATGCTTTT-3′

The primers for this sequence were as follows: forward: ATCTGCATACATGAGGGGCG, reverse: GGAACCGACGAGGTGAACAT. The product lengths were 114 base pairs. The primers hybridize and amplify the region highlighted in green. This target DNA sequence in the promoter region of the PER2 gene was chosen because it contained a canonical CREB protein binding sequence which is highlighted in red.

PER2 promoter 238290198-238290477

5′-AGCCCCGGGAGGCCTGCATGCTGTTCACACACTCAGTCAGGTGGCCCCTCCTCCTGTTCCTCTGACATTGACACCTCGACACACTCCCCGCCCCCTCTCCCTAACACATACACACACAAATGCCAAGCAAACCCAGGCCCGGCTGCTGCGCCCTGCACACACCCAAAGGCTCTTTTGTTTCTTCCCTCCCATTGACGTCAATGGGGAGCTCCATTGTTCTGGAAACAAGAGTAAACAGACAGCTCATCCACACCTTACCGAGATTCTTCTTCATGCTTTT-3′

3′ → 5′

AAAAGCATGAAGAAGAATCTCGGTAAGGTGTGGATGAGCTGTCTGTTTACTCTTGTTTCCAGAACAATGG

AGCTCCCCATTGACGTCAATGGGAGGGAAGAAACAAAAGAGCCTTTGGGTGTGTGCAGGGCGCAGCAGCC

GGGCCTGGGTTTGCTTGGCATTTGTGTGTGTATGTGTTAGGGAGAGGGGGCGGGGAGTGTGTCGAGGTGT

CAATGTCAGAGGAACAGGAGGAGGGGCCACCTGACTGAGTGTGTGAACAGCATGCAGGCCTCCCGGGGCT

5′ → 3′ AGCCCCGGGAGGCCTGCATGCTGTTCACACACTCAGTCAGGTGGCCCCTCCTCCTGTTCCTCTGACATTG

ACACCTCGACACACTCCCCGCCCCCTCTCCCTAACACATACACACACAAATGCCAAGCAAACCCAGGCCC

GGCTGCTGCGCCCTGCACACACCCAAAGGCTCTTTTGTTTCTTCCCTCCCATTGACGTCAATGGGGAGCT

CCATTGTTCTGGAAACAAGAGTAAACAGACAGCTCATCCACACCTTACCGAGATTCTTCTTCATGCTTTT

### 2.6. Chromatin Immunoprecipitation

This assay was performed according to the protocol for the SimpleChIP^®^ Plus Enzymatic Chromatin IP Kit (Agarose Beads) from Cell Signaling Technology (Danvers, MA, USA). The presence of the target gene promoter sequences in both the input DNA and the recovered DNA immunocomplexes was detected by qPCR. 

The primer pairs for specific promoter regions are shown (refer to DNA constructs in Section 2.2). The data obtained were normalized to corresponding DNA input control. A total of 4 µL of indicated antibody was used for each immunoprecipitation (IP). The details of the antibodies used are Anti beta Arrestin 1 antibody (ab31868; Abcam, Cambridge, UK), Anti-GFP antibody—ChIP Grade (ab290; Abcam), Histone H4K16ac (Acetyl Lys16) antibody (GT1271; Invitrogen, Waltham, MA, USA) and Anti-CREB Antibody (PA1-850; Invitrogen).

### 2.7. Nuclear Extraction

Once at 70% confluence, the two groups of cells for nuclear extraction (described previously) were washed with PBS and incubated with 2 mL TrypLE for approximately 5 min. The cells were then resuspended in 5 mL media and centrifuged at 2000 rpm for 3 min and thereafter were similarly resuspended and washed twice with 5 mL PBS. The nuclear fractions were then extracted using the NE-PER Nuclear and Cytoplasmic Extraction kit (Thermo Scientific, Waltham, MA, USA). Nuclear extracts (NE extracts) were harvested in nuclear extract buffer (NE buffer) containing 20 mM NaF and 1 mM phenylmethyl sulphonyl fluoride.

### 2.8. Immunoprecipitation

The NE extracts, which were obtained as described above, were quantified using the Bradford method (Bradford, UK, 1976) with a colorimetric assay Protein Assay (Bio-Rad, Hercules, CA, USA) according to the manufacturer’s instructions. Thereafter, 100 μg of NE extracts was used for each immunoprecipitation reaction. The protein in lysis buffer containing 50 mM Tris-HCl, 150 mM NaCl and 0.5% NP40 with protease inhibitors (Calbiochem, San Diego, CA, USA) was incubated at 4 °C with the indicated antibody buffer overnight (see below). Thereafter, the immunocomplexes were precipitated using Protein G Dynabeads, followed by a 3 h incubation with Protein A Dynabeads (Life Technologies, Waltham, MA, USA) at 4 °C. Immunoprecipitates were washed four times in lysis buffer and subjected to standard Western analysis. Antibodies used in this study were anti-CFP (orb256068; Biorbyt, Cambridge, UK), Anti-Beta Arrestin 1 antibody (ab31868; Abcam) and Anti-Delta opioid receptor antibody (ab 176324; Abcam).

### 2.9. Immunocytochemistry

N/TERT-1 keratinocytes were seeded at 2000 per well on coverslips in a 12-well plate. Upon reaching 70% confluence, these cells were then treated with 100 nM Met-enkephalin for 5 min or with control KSFM media. These cells were then fixed in cold 1:1 methanol and acetone fixative for 5 min. The cells were then washed with 1× PBS, and unspecific binding of antibody was achieved by using CAS-Block™ Histochemical Reagent (catalogue number: 00-8120) from Thermo Scientific. Thereafter, the cells were treated with primary antibodies for Chicken anti-GFP (Abcam, ab13970) (to target GFP-tagged DOPr) and Rabbit anti-β-arrestin 1 overnight at 1:5000 dilution in blocking buffer. The secondary antibodies used were goat anti-chicken which was conjugated to a fluorophore sensitive to a light of wavelength 488 nM (Abcam, ab150169) and donkey anti-Rabbit which was conjugated to a fluorophore sensitive to a light of wavelength 594 nM (Abcam, ab176324) at 1:2000 dilution in 1× PBS.

### 2.10. SDS-Polyacrylamide Gel Electrophoresis (SDS-PAGE)

A total of 20 μg of protein per cell extract was separated on SDS-polyacrylamide gels (Laemmli, 1970). Samples were prepared by addition of 6× SDS sample buffer and heating at 95 °C for 5 min. The acrylamide concentration was adjusted to 10% in a casting gel buffer. The composition of the stacking gel was 5% acrylamide in stacking gel buffer. Electrophoresis was performed with SDS running buffer at 30 mA per gel using MiniPROTEAN^®^ electrophoresis chambers (Bio-Rad). The protein ladder used as a reference was the Precision Plus Protein™ Dual Color Standards (#1610374; Bio-Rad).

### 2.11. Immunoblot Analysis (Western Blot)

After protein separation via SDS-PAGE, samples were subsequently transferred to a nitrocellulose membrane (Bio-Rad). Blotting was performed at 100 V for 1 h 32 min with the Mini Trans-Blot^®^ Cell (BioRad). Membranes were next incubated for at least 1h in TBS (20 mM TRIS, 150 mM NaCl, pH 7.5) blocking solution, containing 0.1% Tween-20 and 5% non-fat milk. The indicated primary antibody was added to 5% Bovine serum Albumin (BSA) (A4737; Sigma-Aldrich) at a dilution of 1:2000 and incubated with the membrane at 4 °C overnight. Membranes were washed 3–5 times for at least 5 min in the aforementioned blocking solution and then incubated with the indicated secondary antibody and 1 μL of StrepTactin-HRP conjugate for 1 h at room temperature. After a final wash for 5 min with blocking solution, the blot was developed with a suitable development reagent. 

### 2.12. Ligand Binding Assay

Met-enkephalin conjugated to the fluorophore TAMRA (Metenk-TAMRA) was used to detect the expression of DOPr on N/TERT-1 keratinocytes. Cells were seeded in triplicates in 6-well plates and grown to 70% confluence in K-SFM supplemented with BPE, EGF and 0.4 mM Ca^2+^ media. On the day of the experiments, BPE- and EGF-free K-SFM was used. Cells were either treated with 1 μM dexamethasone for 1 h or left untreated. Thereafter, both groups were trypsinized, washed and resuspended in 500 μL of 1X PBS (Ca^2+^, Mg^2+^-free). The cells were then incubated on ice with 100 nM Metenk-TAMRA or left untreated for the unstained control for 20 min, followed by sorting in the BD FACS CantoTM II. The observed cell populations were gated to isolate TAMRA-positive cells. Analysis was carried out by gating the live cell population in unstained/untreated cells. Further gating based on this selected population helped quantify Metenk-TAMRA positive cells in the dexamethasone-treated and -untreated cells. This analysis was carried out using the FlowJo^TM^ v10.4 software. Obtained population numbers were recorded and then plotted using GraphPad Prism5 (GraphPad Software Inc., San Diego, CA, USA).

### 2.13. Statistical Analysis

Results for 3 experiments are shown as means ± standard error mean (SEM). The statistical significance of differences between the control group and treatment groups was determined by a two-way ANOVA and Bonferroni’s post hoc comparison. A 5% level of probability was considered significant.

### 2.14. Cosinor Analysis

To analyze rhythmicity in expression, a single-component cosinor model $RQ\left(t\right)=M+Acos\left(\frac{2\pi }{P}t+\phi \right)$ was fitted to the expression profiles obtained from the qPCR data. M represents the midline statistic of rhythm (mesor), A the amplitude and P the period of oscillation, fixed at 24 h. The angle ϕ is the acrophase, the time at which the maximum RQ occurs in each cycle. For each gene, regression of the model against experimental data yields estimates of M, A, φ and associated statistics. The 5 h time point was omitted from the analysis as its inclusion did not yield a proper fit of the cosinor model to the *PER2* control data. The reason may be the strong induction of *PER2* expression caused by dexamethasone [24].

## 3. Results

### 3.1. N/TERT-1 Keratinocytes Exhibit a Robust Circadian Rhythm upon Synchronization with 1 µM Dexamethasone

The applicability of N/TERT-1 keratinocytes for studies on the circadian rhythm was assessed by examining the circadian rhythm of the core clock genes *PER2* and *BMAL1*. It was previously reported that the expression of *BMAL1* peaked at 5–10 h and *PER2* peaked at 25 h post synchronization with dexamethasone in HEKs [25]. In keeping with this finding, N/TERT-1 keratinocytes in our study exhibit a similar and successive pattern in the expression of *BMAL1* and *PER2* (Figure 1A,B). The peak expression of *PER2* mRNA was observed at 15–20 h post synchronization. *BMAL1* was the first core clock gene to peak 10 h post synchronization. The time between two successive peaks in *BMAL1* expression was 25 h, which approximately corresponds to a circadian rhythm. These results also indicated that dexamethasone, a glucocorticoid, is a potent synchronizer of N/TERT-1 keratinocytes, just as in fibroblasts [26].

### 3.2. Met-Enkephalin Treatment Induces a Phase Shift in PER2 Expression 

We next sought to assess the effect of the DOPr agonist Met-enkephalin on the various parameters that define the circadian rhythm. To this end, we subjected the obtained gene expression profiles upon Met-ENK treatment (Figure 2A) to routinely used cosinor analysis. 

Figure 2B shows the mathematical fit to the *PER2* and *BMAL1* expression profiles from the control and the Met-ENK-treated cells. All expression profiles cycle rhythmically. While the mesors and amplitudes are not significantly affected by the treatment, the mean acrophase in the *PER2* expression profiles shows a statistically significant 5.6 h delay following Met-ENK treatment vs. control (Table 1). For *BMAL1*, an acrophase shift of 1.7 h is obtained, but this difference is not statistically significant. Figure 2B(b,d) provide a graphical representation of the elliptical 95% confidence regions for the amplitude-acrophase pairs estimated from the cosinor analysis, showing the significant shift in peak *PER2* expression caused by Met-ENK.

### 3.3. Met-Enkephalin Treatment Induces a Change in Clock-Controlled Gene (CCG) Expressions

It has been shown that Dec2 (BHLHE41/Sharp1), DBP (D site of albumin promoter (albumin D-box) binding protein) and Tef (Thyrotroph Embryonic Factor) belong to the PAR bZIP (Proline and Acidic amino acid-Rich basic leucine ZIPper) family of protein and that they contain E-box sequences in their promoter regions [27,28]. These sequences are targets for the binding of the *BMAL1* and Clock heterodimer which then induce the rhythmic expression of their target genes. *PER2* is known to bind to this heterodimer and thus inhibit the expression of the target genes *DEC2, DBP* and *TEF*. Hence, we hypothesized that if the phase shift in *PER2* expression was not a stochastic effect, it would affect the expression of the above-mentioned genes. We found that the phase shift in *PER2* gene expression does indeed induce significant changes in *DBP* and *TEF* expression at the 30th hour and 40th hour post synchronization. However, no significant changes were observed in *DEC2* expression (Figure 3).

### 3.4. Met-Enkephalin Treatment Induced Activation of DOPr Results in Internalization and Nuclear Co-Localization of DOPr and βarr1

In an attempt to decipher the molecular mechanisms involved in inducing a phase shift in *PER2* expression, we overexpressed the DOPr-tagged with GFP to visualize the change in the pattern of expression of DOPr upon Met-ENK treatment. We observed that the DOPr moved from being localized at the periphery of the cell to being internalized and localized into the perinuclear area in the cytoplasm. This condition could be observed from the 5th hour of the experiment until the 45th hour of the experiment (Figure 4A).

To test the hypothesis that βarr1 undergoes nuclear localization, we treated the cells with 100 nM Met-ENK for 5 min. We found that along with βarr1, even the DOPr undergoes nuclear localization (refer to Figure 4A,B). These results were further confirmed when the nuclear fractions of cells overexpressing CFP-tagged βarr1 and GFP-tagged DOPr were subjected to a CFP pulldown using an anti-CFP antibody. The pulldown lysate was then subjected to a Western blot analysis and tested for the presence of DOPr. The blot was then quantified to measure the increase in the amount of DOPr and βarr1 present in the nucleus as a result of Met-ENK treatment (Figure 4C).

### 3.5. Met-Enkephalin Treatment Enhances and Induces a Phase Shift in Rhythmical βarr1 Binding on the PER2 Promoter and Enhances βarr1 Binding to Acetylated H4 in the PER2 Promoter

In addition to undergoing nuclear localization, βarr1 was reported to bind and acetylate histone protein H4 within target genes’ promoters [8]. To test our hypothesis that since βarr1 binding to the *PER2* promoter peaks at 25 h post synchronization, then in order to facilitate the induction in *PER2* expression, it must bind to CREB and acetylate histone 4 in the *PER2* promoter, we carried out ChIP experiments wherein we pulled down βarr1 from chromatin lysates and subsequently carried out a real-time quantitative PCR to compare the amounts of βarr1 bound to the *PER2* promoter in control and Met-ENK-treated N/TERT-1 keratinocytes over a 45 h period. We found that βarr1 binding in control cells peaked at the 15–20th hour, and in Met-ENK-treated cells, it peaked at the 20–25th hour. We then carried out re-ChIP experiments with cells synchronized for 25h with or without Met-ENK treatment. We similarly pulled down βarr1 and then subsequently pulled down CREB and acetylated histone 4 (H4K16ac) using anti-β-arrestin1, anti-CREB and anti-H4K16ac antibodies. Upon doing so, we found that there was a concomitant increase in the binding of CREB to the *PER2* promoter as well as an increase in H4 acetylation. 

### 3.6. DOPr Expression Is Essential for Maintaining Rhythmicity in PER2 Expression

We initially knocked down the DOPr to test whether the phase shift seen in *PER2* expression as a result of Met-ENK treatment was DOPr-mediated. We found that DOPr knockdown cells (DOPr KD) did not exhibit rhythmicity in *PER2* expression (Figure 5A–C). This finding, together with the findings that (i) both DOPr and βarr1 localize to the nucleus (Figure 4A–E) and (ii) βarr1 and DOPr bind to the *PER2* promoter (Figure 6), led us to the conclusion that DOPr is essential for maintaining rhythmicity in *PER2* expression.

### 3.7. DOPr Expression Is Not Affected by Dexamethasone Treatment

A ligand binding assay was conducted to detect the variation in receptor expression caused by dexamethasone-mediated circadian synchronization using Metenk conjugated to fluorophore TAMRA (Metenk-TAMRA). The binding of Metenk-TAMRA indicates that approximately 3.8% of the gated live N/TERT-1 population expresses DOPr. This expression does not undergo significant variation upon treatment with dexamethasone (Figure 7).

## 4. Discussion

DOPr is predominantly expressed in the more differentiated layers of the human epidermis [6] which correlates well with the expression of the endogenous ligand enkephalin [11] at that level. DOPr has also been seen to be highly expressed in the epidermal component of human hair follicles, having a hair-growth-promoting effect [29]. In the central nervous system, DOPr maintains systemic homeostasis [30] and modulates circadian activity [31]. Upon the discovery of DOPr in skin and its similar role in maintaining skin homeostasis [32,33,34,35,36,37,38], we hypothesized that DOPr activation could have an effect on the circadian rhythm in skin.

At the molecular level, there are no observed differences in the circadian clock of neurons in the central nervous system and peripheral cells [39]. One arm of the circadian rhythm feedback loop involves BMAL1 and CLOCK proteins. These proteins dimerize and interact with the E-box elements of their target genes, such as *Cry* and *Per*, which form the other arm of the feedback loop. The CRY and PER proteins then heterodimerize and undergo phosphorylation in the cytoplasm. They are then subsequently translocated to the nucleus, where they disrupt the BMAL1-CLOCK heterodimer, thereby repressing their own transcription [40]. Dexamethasone is well known to be a powerful synchronizer of the circadian rhythm [26]. Therefore, it is not surprising that N/TERT-1 keratinocytes exhibited a robust circadian rhythm upon synchronization with 1 µM dexamethasone. However, this is the first time such a phenomenon has been demonstrated in a mammalian epithelial cell line.

We wondered how the expression of circadian clock genes *BMAL1* and *PER2* is affected by the activation of the DOPr by the endogenous ligand Met-enkephalin. We found that the activation of DOPr resulted in a phase shift in *PER2* expression as confirmed by cosinor analysis. Activating the DOPr system via Met-ENK caused a phase delay, which was quantified as 5.6 h. This is notable as it represents a more prominent delay than that seen with glucocorticoids [24]. To our knowledge, we are the first to use DOPr to manipulate epithelial cells such as keratinocytes. Activation of DOPr also resulted in changes in the expression of clock-controlled genes (CCGs) such as *DBP, DEC2* and *TEF*. These CCGs are regulated by various transcription factors, such as the proline- and acid-rich basic region leucine zipper (PAR-bZIP) family members DBP (D site of albumin promoter-binding protein), TEF (Thyrotroph embryonic factor) and HLF (hepatic leukemia factor), the basic leucine zipper (bZip) protein E4BP4 and the basic helix-loop-helix proteins (bHLH) DEC1 and DEC2 (BHLHE40, BHLHE41). The CLOCK/BMAL1 heterodimer controls the expression of these transcription factors [40,41,42]. PAR-bZip family members have essential roles in neurotransmitter metabolism, xenobiotic detoxification and immune functions. DBP regulates detoxification and drug metabolism enzymes, while TEF activates gene expression in developing thyrotrophs. DEC2 represses the expression of orexin, thereby affecting arousal and wakefulness [41]. DEC2 interacts with *BMAL1* and inhibits the transactivation of *PER1* [43,44]. The phase shift in *PER2* gene expression, by the activation of DOPr, does indeed induce significant changes in *DBP* and *TEF* expression at the 30th hour and 40th hour post synchronization. The fact that the amplitude is increased and phase is delayed by the activation of the DOPr indicates that the opioid system in epidermal keratinocytes influences the circadian rhythm, especially in terms of the cell cycle and metabolism of the skin, where *PER2* is already known to play a critical role.

We then looked to the overexpression system for clues that would lead us to the molecular mechanisms responsible for this induction in phase shift in *PER2* expression. We found that the activation of the receptor using Met-ENK results in the internalization of the receptor from the 5th hour of synchronization until the end of the experiment or the 45th hour. We were able to conclusively show that DOPr can translocate to the nucleus upon activation and co-localize with β-arr1. DOPr activity is known to be regulated by β-arrestin [45]. Interestingly, the tolerance induced by DOPr agonists that induce DOPr internalization with high efficacy has been linked to DOPr recruitment of β-arr1 [46] leading to clathrin-mediated endocytosis of activated and phosphorylated receptors and eventual degradation and/or recycling of the receptor back to the surface [47]. Kang et al. also showed activation of DOPr to trigger the nuclear localization of β-arr1. 

From our experiments, it is apparent that the absence or knockdown of the receptor renders keratinocytes arrhythmic in the expression of their *PER2* gene. The possibility that the absence or lowered expression of DOPr can result in arrhythmicity or reduced rhythmicity can also exist in the case of cells that regulate systemic circadian rhythms via the CNS especially since these receptors have been shown to have constitutional basal activity [48]. We were able to show that activation of the DOPr leads to a phase shift in *PER2* expression via the β-arrestin pathway, the molecular mechanisms for which require that both βarr1 and DOPr translocate to the nucleus. The fact that we observed a concomitant increase in the binding of CREB to the *PER2* promoter, as well as an increase in H4 acetylation, indicates that by binding to the *PER2* promoter, CREB can either enhance or inhibit the process by which the *PER2* gene is transcribed into RNA molecules. This modulates *PER2* gene expression and potentially leads to alterations in its transcription and subsequent keratinocyte and skin circadian rhythm and related functions such as proliferation, differentiation and cell cycles. Therefore, our study and results could have an impact on the way we look at wound healing, cancer, pain, addiction and stress as they are modulated by both DOPr and the circadian rhythm. 

Wound healing and cancer are two sides of the same coin. Cancer, like wound healing, is characterized by migration, adhesion and differentiation. In cases of cancer, it has been observed that the expression of various core circadian genes such as *PER2* is downregulated [49]. The loss of *PER2* expression is closely associated with the genesis and development of oral squamous cell carcinoma [21]. Knockdown of the circadian clock gene *PER2* in human thyroid follicular cells induced AP-1 activity via JNK MAPK signaling activation, which increases cell proliferation [50]. MicroRNA miR-3187-3p promotes the capacity of migration and invasion of head and neck squamous cell carcinomas (HNSCCs) by targeting *PER2* [20]. 

In the context of cancer, a downregulation of factors promoting adhesion between cells and an upregulation of factors promoting migration of cells lead to metastasis, whereas in the case of wound healing, a fine balance between the adhesion and migration of cells serves as a homeostatic mechanism that restores the integrity of the organ [51]. *PER2* has been shown to regulate migration, adhesion of fibroblasts [52] and differentiation of keratinocytes [25]. *PER2* was also found to regulate the expression of NONO which regulates the senescence of myofibroblasts which is essential for wound healing [53,54]. Moreover, mice with mutations to the *PER2* minimal upstream open reading frame showed delayed wound healing and perturbed ability to follow simulated body temperature cycles [55]. Likewise, *BMAL1* regulates the repair and replication of keratinocyte DNA [3], and its activity, according to the molecular circuitry model of circadian rhythms, is influenced by *PER2*. Thus, DOPr can be a potential target for further study into the treatment of cancers related to the dysregulation of *PER2.*

## 5. Conclusions

This investigation elucidates the influence of delta opioid receptor activation on the circadian rhythm in skin. Initially, we charted the intrinsic circadian clock of keratinocytes, focusing on core circadian gene activity. Activation of keratinocytes’ delta opioid receptors with Met-enkephalin induced a phase shift in the expression of the *PER2* gene and affected the expression of the clock-controlled genes *DBP* and *Tef*. Subsequent analysis revealed nuclear translocation of both DOPr and beta-arrestin post-DOPr activation. DOPr knockdown experiments demonstrated that without DOPr, keratinocytes failed to sustain PER2 rhythmicity, underscoring the receptor’s critical role in the circadian system. Moreover, dexamethasone’s lack of effect on DOPr expression suggests that the receptor’s role in circadian modulation is independent of synchronization mechanisms. This study proposes that DOPr might bridge the beta-arrestin pathway and circadian regulation, where beta-arrestin, upon DOPr activation, augments CREB’s interaction with the PER2 promoter and histone H4 acetylation. Highlighting its significance, DOPr emerges as a promising target in the treatment of circadian rhythm disorders, skin cancer and impaired wound healing. 

## Figures and Tables

**Figure 1 cells-13-00232-f001:**
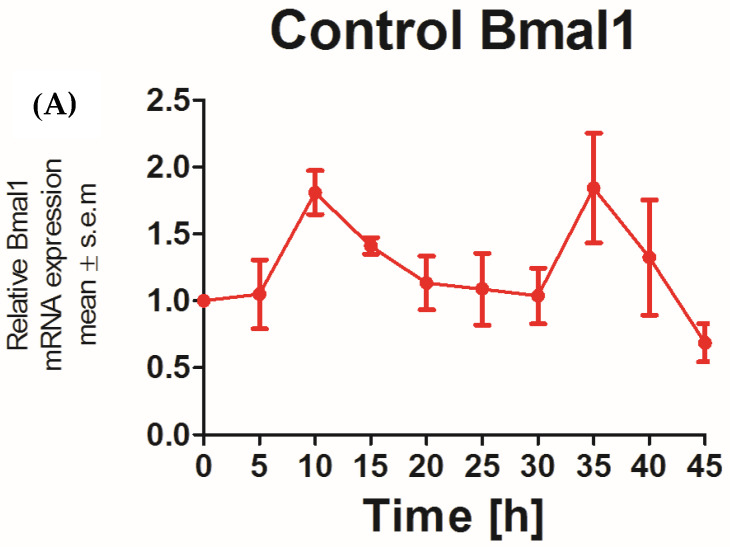
N/TERT keratinocytes exhibit robust circadian rhythms: upon synchronization with 1 μM dexamethasone, N/TERT-1 keratinocytes were synchronized, and gene expression was measured at indicated times. The cells exhibited peak expression in *BMAL1* at 10 h (**A**) and *PER2* at 15–20 h (**B**).

**Figure 2 cells-13-00232-f002:**
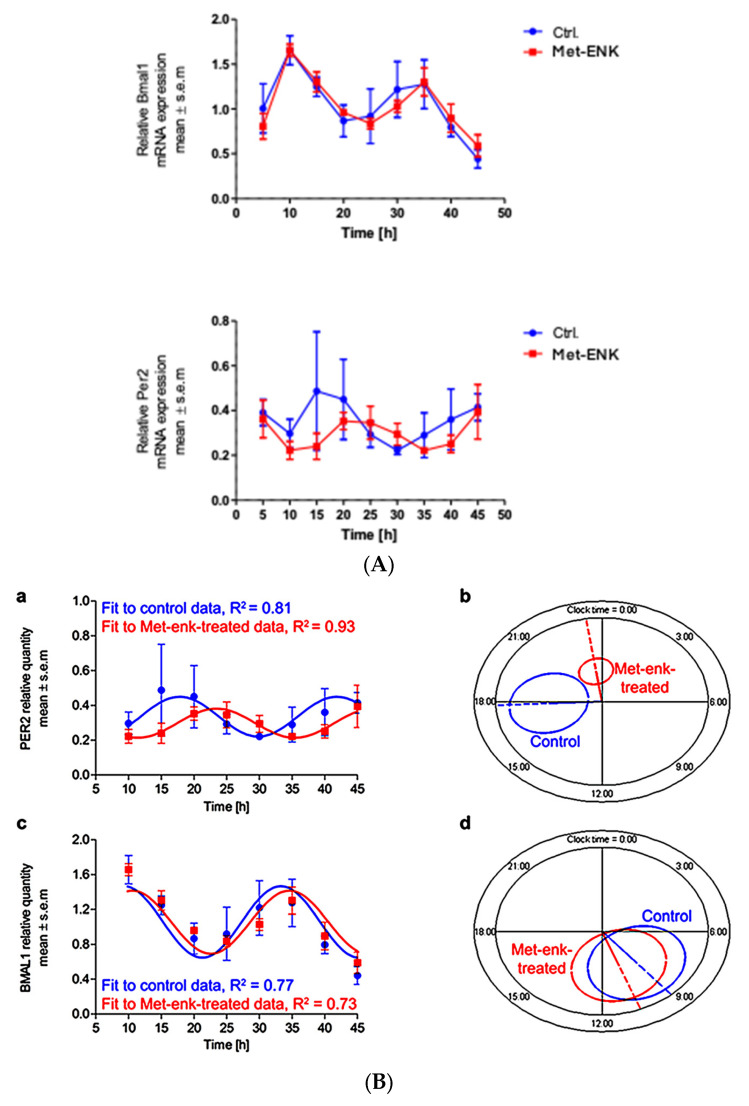
Met-ENK treatment induces internalization of the DOPr and a phase shift in *PER2* expression: (**A**) N/TERT-1 keratinocytes were synchronized, and gene expression profiles were obtained at indicated times. The upper panel and lower panel show expression profiles of *BMAL1* and *PER2* in control cells and cells treated with Met-ENK, respectively. It is apparent that the control group cells exhibited peak expression in *BMAL1* at 10 h and *PER2* at 15–20 h, whereas the Met-ENK-treated group cells exhibited peak expression in *BMAL1* at 5–10 h and *PER2 at* 25 h. *n* = 3 in all panels; results are expressed as mean ± SEM. (**B**) In all cases, the period is fixed at 24 h. The R^2^ coefficients indicate goodness of fit or percentage of rhythm in the data. (**a**) Regression of the cosinor model against the *PER2* mRNA expression profiles (pooled data from *n* = 3 experiments) shows a 5.6 h phase shift in the Met-ENK-treated (red) cells compared to control (blue). (**b**) The 95% confidence regions (ellipses) obtained from the *PER2* regressions are distinct; acrophases φ (clock times indicated by the dashed lines) are significantly different. (**c**) Regression against the *BMAL1* control and Met-ENK treatment data yields no difference in rhythmicity. (**d**) The *BMAL1* 95% confidence regions overlap; A and φ are not significantly different.

**Figure 3 cells-13-00232-f003:**
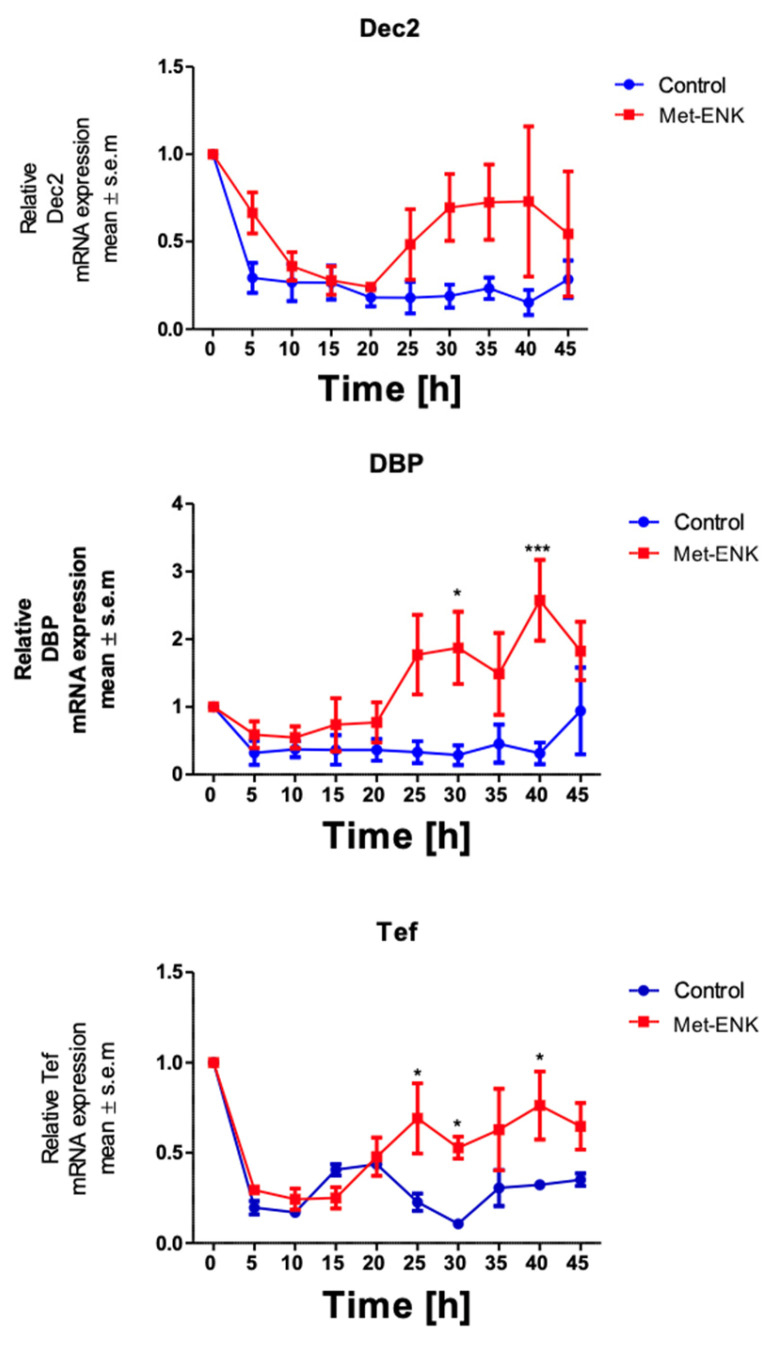
Met-enkephalin treatment induces a change in clock-controlled gene expression: There was no significant change in *Dec2* expression as a result of Met-ENK treatment. However, 25 h post synchronization, the Met-ENK-treated group showed significant changes in *DBP* and *Tef* expression. The data here are the means ± SEM of three independent experiments (*n* = 3). Two-way ANOVA reveals * *p* < 0.05 and *** *p* < 0.001.

**Figure 4 cells-13-00232-f004:**
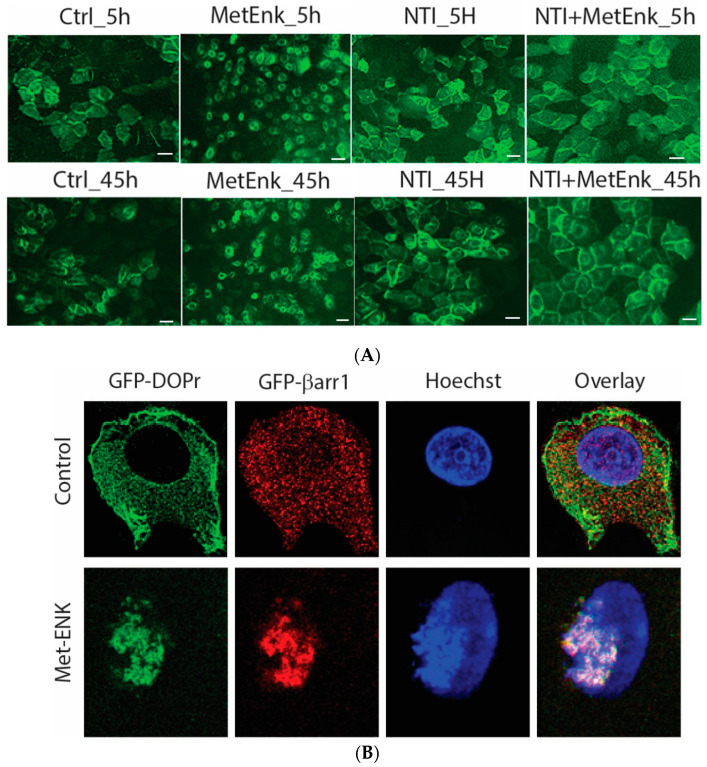
Activation of DOPr leads to internalization and nuclear co-localization of DOPr and βarr1: (**A**) DOPr-GFP overexpressing N/TERT-1 keratinocytes exhibit DOPr to be localized to the cell membrane in control (untreated) cells. This localization remains unaffected in cells treated with DOPr antagonist, i.e., NTI, and combined NTI and agonist (Met-ENK). However, upon Met-ENK treatment, the receptor is internalized and appears to undergo relocalization to the perinuclear region in the cytoplasm. Scale bar 20 μm. (**B**) Confocal visualization of N/TERT-1 keratinocytes overexpressing GFP-DOPr, and CFP-βarr-1 incubated without (Ctrl) or with 100 nM Met-ENK for 5 min before fixation. Upon Met-ENK treatment, both DOPr and βarr1 co-localize in the nucleus. (**C**) The pictures shown are representative of three independents since all blots appeared the same. Values are expressed as the mean ± SD. ** *p* < 0.01. (**D**) N/TERT-1 keratinocytes overexpressing GFP-DOPr and CFP-βarr-1 were incubated without (Ctrl) or with 100 nM Met-ENK for 5 min, and the nuclear extracts were then used for a pulldown using an anti-CFP antibody. The pulldown lysates were then tested for βarr1 and DOPr. (**E**) shows the graph of the semiquantification of DOPr or βarr1. Each bar represents the signal from one representative experiment, *n* = 1.

**Figure 5 cells-13-00232-f005:**
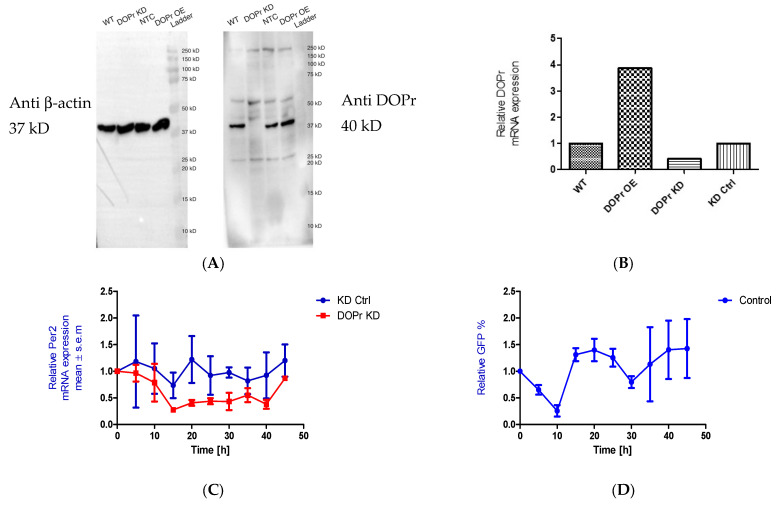
DOPr expression is essential for maintaining rhythmicity in *PER2* expression: (**A**) Constitutive knockdown of the DOPr (KD DOPr) was possible using a lentiviral construct. This was validated by Western blot using an anti-Delta Opioid Receptor antibody (ab176324; Abcam), and a band was seen at the 40 kDa mark when used at a 1/1000 dilution with anti-β-actin antibody (A5441; Sigma-Aldrich) as the loading control and (**B**) quantitative real-time polymerase chain reaction (qPCR), *n* = 1. (**C**) To test for rhythmicity in *PER2* expression, qPCRs were carried out using non-targeting control (KD Ctrl) and DOPr KD cells. It was observed that the DOPr KD cells did not exhibit rhythmicity in *PER2* expression when compared to the KD Ctrl cells which exhibited a trough in *PER2* expression at 15h and 35h post synchronization, *n* = 2. (**D**) Chromatin immunoprecipitation experiments (*n* = 2) targeting the *PER2* promoter in GFP-tagged DOPr overexpressing N/TERT-1 keratinocytes showed that DOPr binds rhythmically to the *PER2* promoter.

**Figure 6 cells-13-00232-f006:**
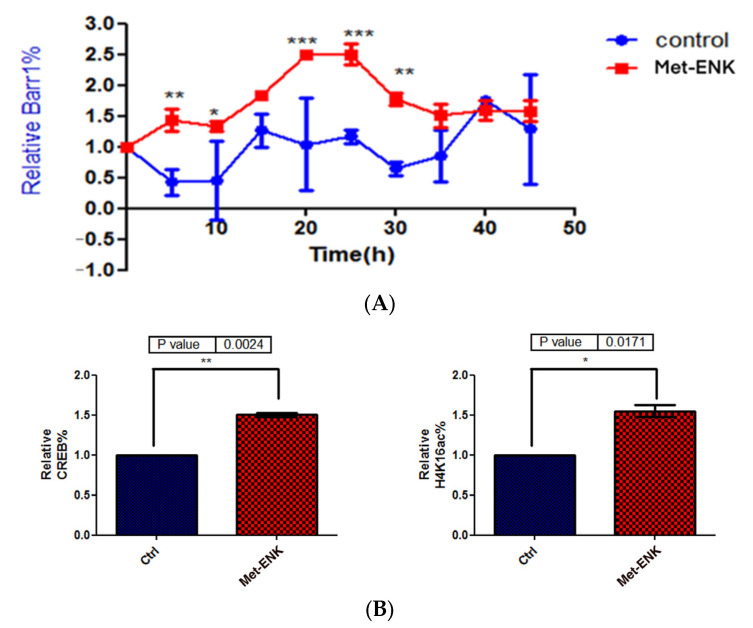
Met-enkephalin treatment enhances and induces a phase shift in rhythmical βarr1 binding on the *PER2* promoter and enhances βarr1 binding to acetylated H4 in the *PER2* promoter: (**A**) Chromatin immunoprecipitation (ChIP) experiments were carried out with anti-βarr1 antibodies, and *PER2* promoter sequences in the input DNA and that recovered from antibody-bound chromatin segments were analyzed by qPCR. The data were normalized to the corresponding input control. All data were then normalized to the corresponding 0 h control. The data shown are the means ± SEM of three independent experiments (*n* = 3). Two-way ANOVA reveals * *p* < 0.05, ** *p* < 0.01, *** *p* < 0.001. (**B**) Re-ChIP experiments were carried out on N/TERT-1 keratinocytes which were synchronized for 25 h with or without Met-ENK treatment. In these experiments, antibodies targeting βarr1 were used. The consequent eluates were then used for pulling down CREB and acetylated histone 4 (H4K16ac), and the presence of the *PER2* promoter sequences in the input DNA and that recovered from antibody-bound chromatin segments was analyzed by qPCR. The data were normalized to the corresponding input control. All data were then normalized to the corresponding 25 h control. The means ± SEM of three independent experiments (*n* = 3) are presented. The data were then subjected to a paired two-tailed *t*-test, and the *p* values obtained for CREB and H4K16ac were 0.0024 and 0.0171, respectively.

**Figure 7 cells-13-00232-f007:**
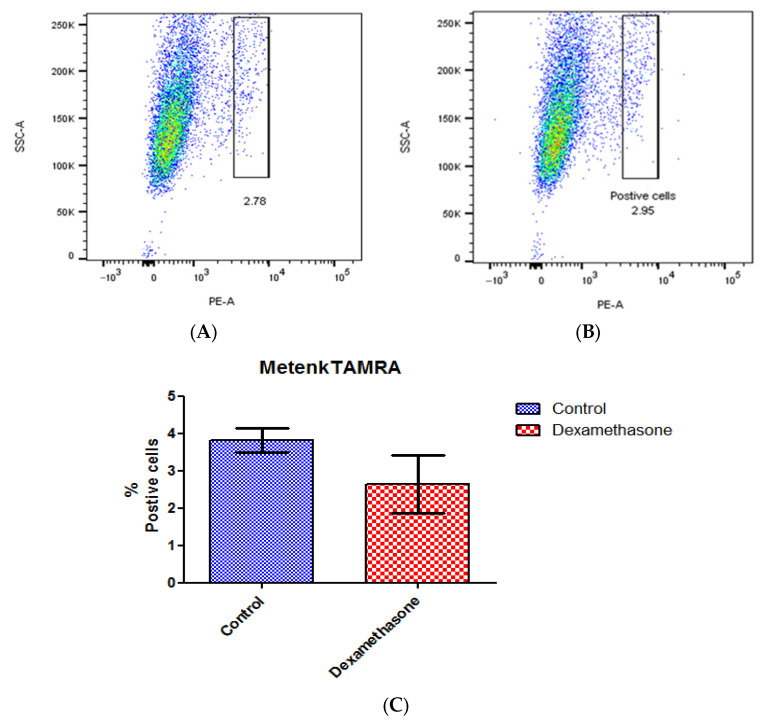
Dexamethasone treatment does not affect DOPr expression in N/TERT-1 keratinocytes. Ligand Metenk-TAMRA binding assay on (**A**) control N/TERT-1 and (**B**) dexamethasone-treated cells identifies DOPr positive population. (**C**) Graphical representation of DOPr positive population percentage. *n* = 4, results are expressed as mean ± SEM.

**Table 1 cells-13-00232-t001:** Results of the cosinor analysis carried out on gene expression profiles of *BMAL1* and *PER2* in control and Met-ENK-treated cells were tabulated. It is apparent that Met-ENK treatment induced a significant difference (*p* = 0.0013) in the acrophase of *PER2* expression.

Data	M[RQ] (Mean ± Std. Error)	A[RQ] (Mean and 95% CI)	f[h] (Mean and 95% CI)	*p*-Value
*PER2* Control	0.34 ± 0.017	0.11 (0.030 to 0.20)	17.8 (15.0 to 21.0)	0.016
*PER2* Met-ENK-treated	0.30 ± 0.0077	0.080 (0.050 to 0.12)	23.4 (21.6 to 25.1)	0.0013
*BMAL1* Control	1.1 ± 0.08	0.41 (0.070 to 0.75)	9.29 (5.34 to 13.3)	0.025
*BMAL1* Met-ENK-treated	1.1 ± 0.08	0.36 (0.030 to 0.70)	10.6 (5.63 to 15.1)	0.038

## Data Availability

The authors confirm that the data supporting the findings of this study are available within the article.

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
