# Peer review of "The Opioid Receptor Influences Circadian Rhythms in Human Keratinocytes through the β-Arrestin Pathway"

_cells, 2024, doi:10.3390/cells13030232_

Round 1
Reviewer 1 Report (Previous Reviewer 3)
Comments and Suggestions for Authors
Authors have revised the manuscript and partly solved the issues.
1) Methods are still confusing. Sequence formatting must be improved. ChIP is still included under qPCR. Subheading 2.11 occurs 5 times. "u" is still used instead of "µ" (a simple find command could be used).
2) The rhythmicity of PER2 and changes after treatment should be further clarified in the manuscript. The demonstration of rhythmicity after MetEnk treatment (fig. 2) and, importantly, in the knockdown experiments should be improved; the error bars in the non-targeting experiment are large with n=2.
3) Author contributions should be double-checked.
Author Response
Thank you for your comments, please see the attached responses.

Reviewer 2 Report (New Reviewer)
Comments and Suggestions for Authors
Comments and Suggestions for Authors
The manuscript entitled “The opioid receptor influences circadian rhythms in human keratinocytes through the β-arrestin pathway” describes that DOPr activation leads to a phase shift in PER2 expression via β-arrestin-1-facilitated chromatin remodeling. Overall, this manuscript is well written. However, there are minor points the authors should revise for clarification and publication.
1. There are some formatting errors and writing errors in this manuscript that need careful checking and correction, e.g.,
a) Please remove the highlighted yellow in the main text (e.g., material & methods 2.2 – page 2, lines 95-97; 2.11; 3.8, etc.).
b) Please write in full sentences for all abbreviations if mentioned for the first time (e.g., K-SFM, BPE- and EGF, page 2, line 93; ChIP, line 113, etc.).
c) Please revise all numbering/subheadings in the Material & Method. Four subheadings use the same number 2.11.
d) The labels for the groups Met-ENK, NTI, NTI+Met-ENK are not standardized. Different labels were used in the figure legend, label, and text.
2. To add information regarding β-arrestin in the introduction.
3. How long were the cells incubated? Is there any media change during the cultures? If yes, please specify the treatment method. So, in total, there are three treatment groups, right?
4. For how long were cells incubated with 2 ml TrypLE? Please specify in the nuclear extraction method.
5. Please check and complete the western blot procedure—no secondary incubation.
6. There should be three treatment groups (Met-ENK, NTI, NTI+Met-ENK), right? Why was only one treatment group shown in most of the results? Suggest including the data for all groups.
7. Suggest separating Table C from Figure 2 as a stand-alone table.
8. The WB images in Figure 4D are not clear. Suggest replacing them with clearer band images.
9. Figure 4E shows the graph of the semi-quantification of DOPr or βarr1. Each bar represents the signal from one representative experiment, n=1. Why only 1? Can you compile all signals and take an average?
10. Suggest combining Figures 5 and 6.
11. For Figure 7, why does the data show only n=1, n=2, and not standardized? How many samples are needed for all groups, and how did you calculate the sample size?
12. For Figure 7A, suggest labeling the size/molecular weight of the protein ladder.
13. Suggest continuing the line or combining lines 480-481 into one paragraph (discussion).
14. Suggest adding an overall summary/conclusion of the study.
15. Suggest citing the most recent publications for references.
Author Response
Thank you for your comments, please see the attached responses.

This manuscript is a resubmission of an earlier submission. The following is a list of the peer review reports and author responses from that submission.
Round 1
Reviewer 1 Report
Comments and Suggestions for Authors
The manuscript confirmed the relationship between cutaneous opioid receptor and genes regulating the circadian rhythm, specifically the core clock gene per2. This regulation resulted to be regulated by βarrestin pathway. The development of the work is sound and logical, however, I have one point which could be more specific:
1- In material and methods 2.2. treatment: writers specified that one day of the experiment, cells were treated for one hour with either dexamethasone or Met-enkephalin. This would lead the reader to assume that all experiments were treated in the same manner, however, in 2.4 and 2.6 of material and methods they stated that they used Met-enkephalin for 5 minutes only before the experiments. Please clarify
The manuscript contained different letter types which made it a little bit crowded, please unify the letter size and font type.
Reviewer 2 Report
Comments and Suggestions for Authors
Major Comments:
1. The authors use dexamethasone to "synchronize" the cells. However, dex will also change the expression of multiple genes via its activation of the glucocorticoid receptor. It is impossible to discern which effects are "circadian" and which are effects of the drug. The effects of dex on DOPr and beta-arresin expression do not appear to be tested.
2. The circadian rhythm is not clear in some of the figures. This is particularly the case for Figure 7C, which is critical to the authors' conclusions. The "control" rhythm is difficult to see in this experiment, and the error bars are very large. The effect is not convincing.
3. Overall, it is hard to envision how knockdown of a single GPCR can affect circadian rhythm to such an extent. The effects of activation of other GPCRS that act via beta-arrestin should be tested.
4. Data should be presented as SD, not SEM.
5. GPCRs are typically internalized fairly rapidly following activation. In this case, the authors do not see "internalization" until after five hours, and do not comment on why it is so slow. They actually seem to be referring to nuclear localization, rather than internalization. When discussing the time course, the author refer to "Figure 2D", but there is no such figure.
Comments on the Quality of English Language
The English quality is adequate.
Reviewer 3 Report
Comments and Suggestions for Authors
Authors have analysed opioid receptors in N/TERT keratinocytes. They found that DOPr activation involved the β-arrestin pathway and chromatin modification and resulted in phase shift of circadian gene PER2. The findings are interesting with regard to the control of PER2 in circadian mechanisms and possibly wound healing and cancers. The manuscript is mostly clear but can be improved in layout and structure.
Detailed comments:
1) The organisation of the methods is confusing. Subheadings are not well assigned, some aspects are rather short and others very detailed. ChIP experiments - section starts in the middle of another paragraph - are included with all sequence details; this can be useful but should be more suitably formatted. Manufacturers and details of antibodies etc should be consistently included. Some details are repeated within the section. 2-DDCT should read 2-ΔΔCt. "micro-" should be abbreviated as "µ", not "u", in Methods and elsewhere.
2) Genes and proteins should be named according to consensus nomenclature, i.e. human gene symbols in italics and capitals, rodent genes with only first letter uppercase.
3) Figures should be re-formatted. Why is 2C so large? Letters for panels could be more suitably sized and positioned.
4) All figures should be introduced in the text; "refer to" is not needed but the importance of each figure should be clarified. Figure 2D is not included.
5) The cosinor analysis should be described with some details.
6) Results are based on rhythmicity of BMA1 and PER2 expression. Is the 24h/25h period clear with regard to the expression of PER2?
7) BMA1 does not seem to show a shift after treatment; this is seen in the expression curve and confirmed in the cosinor analysis. The difference of the peak with and without treatment as stated is not clear.
8) NTI treatment should be introduced.
9) Size bars should be included into microscopy pictures, especially for the confocal imaging.